# Glycosylation of Epigallocatechin Gallate by Engineered Glycoside Hydrolases from *Talaromyces amestolkiae*: Potential Antiproliferative and Neuroprotective Effect of These Molecules

**DOI:** 10.3390/antiox11071325

**Published:** 2022-07-05

**Authors:** Juan A. Méndez-Líter, Ana Pozo-Rodríguez, Enrique Madruga, María Rubert, Andrés G. Santana, Laura I. de Eugenio, Cristina Sánchez, Ana Martínez, Alicia Prieto, María Jesús Martínez

**Affiliations:** 1Centro de Investigaciones Biológicas Margarita Salas, Department of Microbial and Plant Biotechnology, Spanish National Research Council (CSIC), Ramiro de Maeztu 9, 28040 Madrid, Spain; jmendez@cib.csic.es (J.A.M.-L.); ana.pozo@cib.csic.es (A.P.-R.); lidem@cib.csic.es (L.I.d.E.); aliprieto@cib.csic.es (A.P.); 2Centro de Investigaciones Biológicas Margarita Salas, Department of Structural and Chemical Biology, Spanish National Research Council (CSIC), Ramiro de Maeztu 9, 28040 Madrid, Spain; enrique.madruga@cib.csic.es (E.M.); ana.martinez@csic.es (A.M.); 3Department of Biochemistry and Molecular Biology, School of Biology, Instituto de Investigación Hospital 12 de Octubre, Universidad Complutense de Madrid, C/de José Antonio Nováis 12, 28040 Madrid, Spain; mrubert@ucm.es (M.R.); cristina.sanchez@quim.ucm.es (C.S.); 4Department of Bioorganic Chemistry, Instituto de Química Orgánica General, Spanish National Research Council (CSIC), C/Juan de la Cierva 3, 28006 Madrid, Spain; andres.g.santana@csic.es

**Keywords:** fungal enzymes, glycoconjugates, polyphenol, glucose, sophorose, xylose

## Abstract

Glycoside hydrolases (GHs) are enzymes that hydrolyze glycosidic bonds, but some of them can also catalyze the synthesis of glycosides by transglycosylation. However, the yields of this reaction are generally low since the glycosides formed end up being hydrolyzed by these same enzymes. For this reason, mutagenic variants with null or drastically reduced hydrolytic activity have been developed, thus enhancing their synthetic ability. Two mutagenic variants, a glycosynthase engineered from a β-glucosidase (BGL-1-E521G) and a thioglycoligase from a β-xylosidase (BxTW1-E495A), both from the ascomycete *Talaromyces amestolkiae*, were used to synthesize three novel epigallocatechin gallate (EGCG) glycosides. EGCG is a phenolic compound from green tea known for its antioxidant effects and therapeutic benefits, whose glycosylation could increase its bioavailability and improve its bioactive properties. The glycosynthase BGL-1-E521G produced a β-glucoside and a β-sophoroside of EGCG, while the thioglycoligase BxTW1-E495A formed the β-xyloside of EGCG. Glycosylation occurred in the 5″ and 4″ positions of EGCG, respectively. In this work, the reaction conditions for glycosides’ production were optimized, achieving around 90% conversion of EGCG with BGL-1-E521G and 60% with BxTW1-E495A. The glycosylation of EGCG caused a slight loss of its antioxidant capacity but notably increased its solubility (between 23 and 44 times) and, in the case of glucoside, also improved its thermal stability. All three glycosides showed better antiproliferative properties on breast adenocarcinoma cell line MDA-MB-231 than EGCG, and the glucosylated and sophorylated derivatives induced higher neuroprotection, increasing the viability of SH-S5Y5 neurons exposed to okadaic acid.

## 1. Introduction

Glycoside hydrolases (GHs) are a very heterogeneous group of enzymes that act on carbohydrates (currently, 173 families reported in the CAZy database [1]). These proteins are generally known to catalyze the hydrolysis of glycosidic linkages, but under certain conditions, they can also synthesize them by transglycosylation [2]. Although this ability has been known for over 60 years, the full synthetic potential of GHs is still untapped.

Transglycosylation is a useful tool to produce valuable glycoconjugates [3], being an alternative to the complex chemical synthesis, which usually involves the formation of toxic byproducts and displays low regio- and stereoselectivity. However, using wild-type enzymes in synthetic reactions has some weaknesses since the transglycosylation yields are limited, and the reversibility of the reaction can result in hydrolysis of the product [4]. To overcome these drawbacks, the development of engineered glycosidase variants, such as glycosynthases and thioglycoligases, appears as one of the most promising technologies. These enzymes are obtained by replacing one of the two catalytic amino acids in the active center with an inert one, generating variants with null or drastically reduced hydrolytic activity and enhanced synthetic capacity [4]. Replacement of the catalytic acid/base residue generates thioglycoligases, enzymes that need donors with a good leaving group, like dinitrophenyl sugars. In the case of glycosynthases, the replacement affects the residue responsible for the nucleophilic attack, and they require a fluorine-activated donor. Generally, glycosynthases have been used for the synthesis of oligosaccharides of different lengths [5] and thioglycoligases to produce novel *S*-glycosides [3]. However, their ability to generate other types of glycoconjugates has been poorly explored.

Phenolic compounds possess one or more aromatic rings with a variable number of hydroxyl groups. They are widely distributed in nature and can be found as simple molecules like aromatic acids or highly polymerized substances such as catechins or tannins [6]. The demonstrated health benefits of many phenolic compounds have attracted the attention of researchers [7]. Among them, epigallocatechin gallate (EGCG), the major antioxidant species of green tea, is a phenolic compound with a variety of therapeutic benefits, including cardiovascular and neuronal protection, antiproliferative or even antiviral effects, which could be useful for the ongoing SARS-CoV-2 pandemic [8,9,10]. Due to its outstanding properties, the enzymatic glycosylation of EGCG could be an interesting strategy to increase its bioavailability and may improve some of its bioactive effects. Several studies have developed this approach, mainly binding α-glucoses to EGCG [11,12], showing that the antioxidant and biological properties of the glycosylated derivatives depend on the glycosylation site [13,14,15,16]. Moreover, all available studies on EGCG glycosylation have been performed by binding glucose moieties, and the incorporation of alternative carbohydrates may positively affect the bioactive properties of EGCG.

*Talaromyces amestolkiae*, an ascomycetous fungus isolated from cereal wastes, is being studied for its ability to secrete high GHs levels [17], among them an acidophilic β-xylosidase (BxTW1) and a glucotolerant β-glucosidase (BGL-1). These enzymes were converted, by site-directed mutagenesis, into a thioglycoligase and a glucosynthase variant, respectively, which were tested to produce EGCG glycosides [18,19]. In this work, both mutagenic variants (BGL-1-E521G and BxTW1-E495A), expressed in the yeast *Pichia pastoris*, were used to optimize the production of three EGCG glycosides (glucoside, sophoroside, and xyloside). In addition, their water solubility and thermal stability were determined, and their antiproliferative and neuroprotective properties were analyzed.

## 2. Materials and Methods

### 2.1. Enzymatic Assays

BGL-1-E521G and BxTW1-E495A were produced in YEPS medium (20 g/L peptone, 10 g/L yeast extract, 10 g/L sorbitol, and 100 mM potassium phosphate buffer, pH 6). Then, the cultures were filtered, concentrated, dialyzed, and subsequently purified by ion-exchange chromatography, as recently described [18,19]. Total extracellular protein concentration was determined using a Nanodrop spectrophotometer (Thermo Fisher Scientific, Waltham, MA, USA) and confirmed by the bicinchoninic acid assay (BCA) method.

Epigallocatequin gallate (EGCG) was the acceptor molecule in all the enzymatic reactions and was acquired from Zhejiang Yixin Pharmaceutical Co. (Jinhua, China). Donors were α-glucoside fluoride (α-GlcF, prepared as previously reported [20]) or *p*NPX (Sigma-Aldrich, San Luis, MI, USA) when BGL-1-E521G or BxTW1-E495A were used as catalysts, respectively.

The standard glycosylation reaction for BGL-1-E521G contained 20 mg/mL of α-GlcF, 5 mg/mL of EGCG and 1 mg/mL of BGL-1-E521G in acetate buffer 50 mM pH 4, and it was carried out at room temperature for 16 h and 500 rpm. The standard glycosylation reaction for BxTW1-E495A included 10 mg/mL of *p*NPX, 5 mg/mL of EGCG and 1 mg/mL of BxTW1-E495A in acetate buffer 50 mM pH 5.5, and it was performed at 40 °C for 4 h and 500 rpm. All assays included 0.1% BSA to minimize activity loss in the enzymatic reactions [21]

### 2.2. HPLC Analysis of Reactions and Purification of Glycosides

An Agilent 1200 series LC instrument equipped with a ZORBAX Eclipse plus C18 column (Agilent, Santa Clara, CA, USA) was used for all determinations. In order to purify the EGCG glucoside and the sophoroside, the column was first equilibrated in a mix of acetonitrile:H_2_O (13:87) with 0.1% acetic acid at a flow of 2 mL/min, and the reaction products were separated isocratically in 8 min. After isocratic elution, the column was washed for 3 min with acetonitrile:H_2_O (95:5), and the system was finally re-equilibrated to the initial conditions for 4 min. For the EGCG xyloside, the column was first equilibrated in a mix of acetonitrile:H_2_O (10:90) with 0.1% acetic acid at a flow of 2 mL/min. In this case, the reaction products were separated using a gradient from 10 to 30% of acetonitrile in 8 min. Then, the column was washed and re-equilibrated as described above.

In both cases, product peaks were detected by monitoring the absorbance at 270 nm, and quantification was done using a calibration curve of non-glycosylated EGCG. The fractions containing the three different glycosides were collected, lyophilized, and stored at −20 °C, prior to biochemical and bioactivity determinations. Their identity was confirmed by ESI-MS and NMR as previously described [18,19].

### 2.3. Optimization of Glycosylation Reactions Catalyzed by BGL-1-E521G and BxTW1-E495A

A response surface methodology approach was followed to optimize the reaction conditions for the synthesis of EGCG glycosides. Design-Expert^®^ v10.0.1.0 (Stat-Ease Inc., Minneapolis, MN, USA) was the software used to generate the corresponding Box-Behnken design matrixes and for data analysis. The parameters selected for building the model of EGCG glucoside and sophoroside production were: α-GlcF concentration (5–50 mM), EGCG concentration (5–50 mM), enzyme dosage (0.2–1 g/L BGL-1-E521G), and reaction time (1–7 h). In the case of EGCG xyloside, *p*NPX concentration (5–50 mM), EGCG concentration (5–50 mM), enzyme dosage (0.5–2.5 g/L BxTW1-E495A), reaction time (1–5 h), and pH (5.5–6.5) were modeled. All reactions were carried out at 25 °C and 500 rpm. After performing the required reactions, they were analyzed by HPLC, as indicated in Section 2.2, to determine glycoside production and conversion. These data were then introduced into the software to generate a polynomial quadratic equation for each glycoside, showing the effect of the independent variables on the process response along with the expected maximum yields and conversions. The models were validated by performing the reactions in which the highest yields and conversions were predicted.

### 2.4. Antioxidant Activity Assays

The antioxidant activity of EGCG and its glycosides was determined by assessing the reduction of the cation radical of 2,2-azino-bis(3-ethylbenzothiazoline-6-sulfonic acid) diammonium salt (ABTS^•+^) using Trolox (6-hydroxy-2,5,7,8-tetramethylchroman-2-carboxylic acid, Sigma-Aldrich) as reference antioxidant compound [22]. The assay was performed in quadruplicate in 96-well plates. The ABTS^•+^ was obtained after 16 h incubation of 3.5 mM ABTS (Roche) with 1.22 mM potassium persulfate (Sigma-Aldrich) in the dark at room temperature. Solutions of increasing concentrations of Trolox (7.8–1000 μM), EGCG, and its glycosides (1–2000 μM) were prepared in ethanol and distilled water, respectively. Negative controls with only ethanol or distilled water were also included in the assay. Then, 10 μL of each antioxidant was mixed with 115 μL of ABTS^•+^ working solution (previously diluted with ethanol to get an absorbance of ~0.7 at 734 nm). The decrease in ABTS^•+^ absorbance (at 734 nm) was monitored with a SpectraMax^®^ M2 microplate reader (Molecular Devices, San Jose, CA, USA) for 15 min by measuring every minute and was represented as:$${\mathrm{ABTS}}^{•+}\;\mathrm{reduction}\;\left(\%\right)=\left(\frac{{\mathrm{Abs}}_{\mathrm{control}}-{\mathrm{Abs}}_{\mathrm{antioxidant}}}{{\mathrm{Abs}}_{\mathrm{control}}}\right)\times 100$$

The curves of ABTS^•+^ reduction (%) were plotted in SigmaPlot (Stat-Ease, Minneapolis, MN, USA) and used to calculate EC50 (μM) of each compound, which corresponds to the concentration of antioxidant required to reduce the initial absorbance of ABTS^•+^ to 50%. The results were also expressed as Trolox Equivalent Antioxidant Capacity (TEAC), which referred to the concentration of antioxidant (μM) that reduced ABTS^•+^ absorbance as 1 μM of Trolox. The TEAC value was calculated from EC50 of each compound and EC50 of Trolox. The significance between the EC50 and TEAC values of each EGCG glycoside, compared to the aglycon, was determined with a *t*-test in SigmaPlot, considering *n* = 4, the number of experiments and significant differences when *p* < 0.01.

### 2.5. Thermal Stability and Solubility Assays

Thermal stability was studied following the protocol used by González-Alfonso et al. [14], with minor modifications. Briefly, EGCG and its derivatives were dissolved at 500 µM in 10 mM sodium acetate buffer (pH 4), and they were incubated at 70 °C. Aliquots of 20 μL were taken at intervals, centrifuged, and analyzed by HPLC, monitoring the loss of EGCG signals. Data were analyzed in quadruplicate, using Student’s *t*-test to determine if the observed values were significantly different.

So as to determine the solubility of the molecules, saturated solutions of EGCG and its glycosides were prepared in distilled water and incubated for 1 h at room temperature at 500 rpm, as previously described [23]. Then, the solutions were centrifuged, filtered, and analyzed by HPLC to check the concentration of each molecule using a calibration curve made with non-glycosylated EGCG.

### 2.6. Antiproliferative Assays on MDA-MB-231 Cancer Cells

The human non-cancerous mammary epithelial cell line MCF-10A was obtained from the American Type Culture Collection (ATCC) and maintained in Dulbecco’s Modified Eagle’s Medium (DMEM)/Ham’s F-12 (1:1) (Gibco by Thermo Fisher Scientific) supplemented with 5% horse serum (Gibco), 10 µg/mL insulin (SAFC Biosciences, Gillingham, UK), 0.5 µg/mL hydrocortisone (Sigma-Aldrich), 20 ng/mL epidermal growth factor (EGF), 10 µg/mL cholera toxin (Sigma-Aldrich), and 50 U/mL of a penicillin-streptomycin solution (Lonza, Basel, Switzerland). The human triple-negative breast adenocarcinoma cell line MDA-MB-231 was obtained from the ATCC and cultured in DMEM supplemented with 10% fetal bovine serum (FBS, Gibco) and 50 U/mL of a penicillin-streptomycin solution (Lonza). Both cell lines were maintained at 37 °C in an atmosphere of 5% CO_2_ and validated in the Genomics Core Facility at Alberto Sols Biomedical Research Institute (Madrid, Spain).

MDA-MB-231 and MCF10-A cells were seeded onto 96-well plates at a density of 4 × 10^3^ cells/well and allowed to attach to the plastic surface for 24 h. The medium was then replaced with fresh DMEM or DMEM/Ham’s F-12, respectively, and after 4 h of serum starvation, cells were treated with EGCG or its glycosylated derivates (1, 10, and 75 µM) for 24 h. Dimethyl sulfoxide (DMSO, Sigma-Aldrich) was used as the vehicle. Following treatment, cells were incubated with 0.1% crystal violet (Panreac, Barcelona, Spain) for 20 min in agitation. The plate was then gently washed with water, and the crystals were resuspended in methanol. Cell viability was determined by reading absorbance at 570 nm with a microtiter plate reader (Rayto Life and Analytical Sciences Co., Ltd., Shenzhen, China) and expressed as a percentage versus vehicle-treated cells set at 100%.

### 2.7. Neuroprotection Assays on SH-SY5Y Neuronal Cells

Human neuroblastoma SH-SY5Y cells were grown in DMEM supplemented with 10% FBS and 1% penicillin/streptomycin at 37 °C and 5% CO_2_. SH-SY5Y cells were seeded onto 96-well plates at 4 × 10^4^ cells per well. After 24 h, EGCG or its glycosides were included at the desired concentrations, followed by the addition of okadaic acid (OA) (Sigma Aldrich, Alcobendas, Spain) at 30 nM just 1 h later. This toxin inhibits protein phosphatases and induces tau hyperphosphorylation and neuronal death [24]. Twenty-four hours after the treatment, cells were incubated with 0.5 mg/mL MTT solution for three hours. Then, 96-well plates were spun down at 40,000 rpm for 15 min and, right after, cell media was removed. Formazan crystals were dissolved in 200 µL of DMSO, and UV-absorbance was measured at 595 nm (Varioskan Flash Microplate reader, Thermo Scientific, Waltham, MA, USA). Stocks of EGCG derivatives were dissolved at 10 mM in water. Data were normalized to the values of the control wells.

## 3. Results and Discussion

### 3.1. Analysis of EGCG Transglycosylation

The rational design of GHs is a very interesting approach for obtaining high added value glycoconjugates. In previous works, we have reported that the glycosynthase BGL-1-E521G synthetizes two glycosides from EGCG and α-GlcF [19] and that the thioglycoligase BxTW1-E495A produces a single EGCG xyloside using *p*NPX as sugar donor [18]. The structures of these three glycosides were fully defined by NMR spectroscopy and mass spectrometry (Figure 1).

These glycosides have two main particularities. First, the sugars are bound to the galloyl group of EGCG, which is a much less studied part of the molecule. Second, to the best of our knowledge, the above works constituted the first reports of the synthesis of EGCG sophorosides or xylosides, since previous studies about EGCG glycosylation described the binding of single glucose or maltose motifs [11,12,13,14,25,26].

The enzyme BGL-1-E521G showed strict regioselectivity on the 5″ position of EGCG, giving epigallocatechin gallate 5″-O-β-d-glucopyranoside as the major transglycosylation product and a minor amount of epigallocatechin gallate 5″-O-β-d-sophoropyranoside [19]. On the other hand, BxTW1-E495A was also extremely regioselective, producing only epigallocatechin gallate 4″-O-β-d-xylopiranoside [18].

Although each glycosylation product must be studied separately, it has been reported that the 4′ and 7 positions are very important in maintaining the antioxidant activity of EGCG [11,15,16]. In fact, the same works showed that a glycosylated form of EGCG with glucoses bound simultaneously to 4′ and 4″ positions dramatically reduced the antioxidant activity of EGCG.

### 3.2. Optimization of Glycoside Synthesis by Response Surface Methodology

Before studying the effect of glycosylation on the solubility, stability and bioactivity of the EGCG derivatives obtained in this work, their production was optimized using a Box-Behnken design response surface method. The parameters modeled were donor and acceptor concentrations, enzyme dosage, and reaction time. In the case of the xyloside, pH was also studied, as the thioglycoligase was found to be highly dependent on the pH of each reaction and the pKa of the groups that it glycosylates [18]. The production and conversion of EGCG glycosides were the parameters selected for optimization. For EGCG glucoside and sophoroside, the matrix of experiments generated by the Design Expert software comprised 29 reactions, while for EGCG xyloside, it contained 46 reactions, as pH was included as an additional variable.

The results of productions and conversions obtained in these reactions were used to generate a polynomial quadratic model for each EGCG glycoside, whose equations can be found in the Appendix A. The analysis of the variance test performed by the software validated the experimental setups.

For the glucoside, the model describes that the highest productions are obtained using high donor and acceptor concentrations, which is in agreement with the results previously reported for other β-glucosidases [23,27]. Regarding the production of the sophoroside, synthesis is favored at low EGCG and high α-GlcF concentrations. Thus, when there is no EGCG to be glycosylated at position 5″, but α-GlcF remains in the reaction mixture, this very regioselective glycosynthase adds a second glucose molecule to the *O*-2 position of the one already incorporated in the glycoconjugate, forming the sophoroside, instead of glycosylating another hydroxyl group from EGCG. Finally, the highest xyloside productions are achieved with elevated donor and moderate acceptor concentrations. In this sense, we observed that the maximum set for our model (50 mM of EGCG) had a negative effect on the reaction yields, which were low regardless of the values of the other variables. Therefore, maximum EGCG concentrations might be inhibiting the thioglycoligase. For all the glycosides, the best conversions relative to initial acceptor concentrations were generally observed using high donors’ concentrations and EGCG in low amounts.

The models were then used to predict the experimental conditions for maximal production and conversions of each EGCG glycoside within the selected limits for the parameters. For the EGCG-glucoside (Figure 2A), the maximum production (13.05 mM) was obtained in 225 min, with 49.8 mM EGCG, 49.8 mM α-GlcF, and 0.93 g/L BGL-1-E521G. The maximum conversion of EGCG for the synthesis of this molecule (Figure 2B) was 44.2% in reactions of 283 min with 5.14 mM EGCG, 49.75 mM α-GlcF, and 0.96 g/L BGL-1-E521G. On the other hand, the maximum production of the EGCG-sophoroside (Figure 2C) amounted to 4.33 mM, which was obtained in 360 min, with 5.12 mM EGCG, 50 mM α-GlcF, and 0.8 g/L BGL-1-E521G. In this case, the maximum conversion of EGCG was 64.3% (Figure 2D), in a reaction of 302 min, with 5 mM EGCG, 49.75 mM α-GlcF, and 0.98 g/L BGL-1-E521G. It should be noted that, as the glucoside and sophoroside are generated simultaneously in the reaction, the total conversion of EGCG is above 90%.

Finally, for the EGCG-xyloside, the maximum production detected (Figure 3A) was 11.9 mM, and it was obtained in a 300 min reaction, using 31.7 mM EGCG as acceptor, 49.8 mM *p*NPX as donor, 2.5 g/L BxTW1-E495A, and 5.65 pH. The maximum conversion of EGCG for the synthesis of this molecule (Figure 3B) was 61.7%, in a reaction of 293 min, with 5.7 mM EGCG, 50 mM *p*NPX, 2.45 g/L BxTW1-E495A, and 5.7 pH.

The use of this multiparametric approach was successful since EGCG conversion for all novel glycosides increased approximately twofold with respect to previous non-optimized conditions (48.8% to 90% for EGCG using BGL-1-E521G and 30% to 60% with BxTW1-E495A) [18,19]. Our data are among the best reported so far in the literature. In general, the use of sucrose phosphorylases, glucansucrases, or cyclodextrin glucanotransferases for the α-glucosylation of EGCG, returns conversion rates between 30 and 70% [11,12,26]. More recently, Kim et al. [13] reported a glucosylation yield of 90% using a dextransucrase, which was like BGL-1-E521G efficiency. To summarize, the rational design of GHs opens new avenues that may be effective in glycosylating this polyphenol with high yields.

### 3.3. Effect of Glycosylation on EGCG Antioxidant Properties

After optimizing their production, the biochemical properties of the novel glycosides were tested. Therefore, the antioxidant activity of EGCG and its glycosylated derivatives was determined by analyzing the reduction of ABTS^•+^. Table 1 shows the Equivalent Antioxidant Capacity (TEAC) values of the different molecules. The TEAC values obtained for all of them are below 1, meaning that they show higher antioxidant capacity than Trolox. However, the data also revealed statistically significant reductions in the antioxidant activity of EGCG glycosides compared to the non-glycosylated compound. The antioxidant capacity of phenolic compounds such as EGCG depends on the presence of free hydroxyl groups, which act as hydrogen donors to scavenge reactive oxygen free radicals. Therefore, it is expected that the glycosylation of these compounds causes a worsening of their antioxidant capacity, since some of the OH of the aglycon have disappeared when the glycosidic bonds are formed [28,29]. This activity loss was also reported in the glycosides of other phenolic compounds such as phloretin [30], resveratrol [31], and gallic acid [32], as well as EGCG [14]. In this sense, using the same method of radical scavenging capacity with ABTS^•+^, it has been reported that EGCG 3′-O-α-d-glucopyranoside and EGCG 7-O-α-d-glucopyranoside experimented a similar reduction of antioxidant activity compared to the non-glycosylated EGCG (1.19 and 2.96-fold, respectively). Thus, the different contributions of the various hydroxyl groups to the antioxidant capacity of EGCG should be considered when glycosylating [13,25].

Moreover, the decrease in EGCG antioxidant capacity is slightly larger for the xyloside than for the glucoside and sophoroside, which may indicate that the 4″ position is more important for scavenging activity than the 5″. Comparison of this result with those reported by Nanjo et al. [16] for a doubly glycosylated EGCG (both 4′ and 4″ hydroxyl groups) with dramatically reduced antioxidant activity allows us to conclude that the single glycosylation in 4″ position does not induce an enormous loss of antioxidant power. Nevertheless, it would be interesting to investigate whether xylose by itself may have some effect on this phenomenon. Regarding the EGCG glucoside and sophoroside, both showed very similar antioxidant properties, which could indicate that the length of this carbohydrate chain does not influence the antioxidant capacities since, in this case, two different glycosides are occupying the same hydroxyl group, rendering similar antioxidant properties.

This work adds more information about the influence of a β-glycosyl moiety in position 4″ and 5″ of EGCG (Figure 1). However, it is still unknown if the type of carbohydrate or the linkage (α- or β-glycosidic) could be influencing these differences. Nevertheless, it should be noted that, despite the loss of EGCG antioxidant capacity after glycosylation, initial levels would probably be restored upon in vivo glycoside hydrolysis when free EGCG is released [33,34].

### 3.4. Thermal Stability of the Novel EGCG Glycosides

The lack of thermal stability of EGCG is probably one of its main limitations when used as a food supplement for humans. As recently discussed by González-Alfonso et al. [14], EGCG could suffer oxidative damage or epimerization processes during its operation under experimental conditions. Therefore, it is important to test if glycosylation of EGCG could modify its thermal stability.

EGCG and its three glycosylated derivatives were incubated at 70 °C, as described in the materials and methods section. The degradation of these compounds was analyzed by monitoring the decrease of their signals by HPLC under the tested conditions. All the results were statistically significant (*p* < 0.05). The data displayed in Figure 4 revealed that EGCG and EGCG-sophoroside have very similar thermal resistance, while EGCG-glucoside showed a clear improvement (14%) in its resistance to degradation compared to EGCG after 3 h of incubation. Finally, the EGCG-xyloside presented reduced thermal stability compared with the non-glycosylated compound.

A recent work reported a slight improvement in thermal stability of epigallocatechin gallate 7-O-α-d-glucopyranoside and a considerable worsening in epigallocatechin gallate 3′-O-α-d-glucopyranoside versus that of EGCG [14]. Another interesting work from Noguchi et al. [35] described that glycosylation of the 5 position of EGCG improved stability by 50%, the highest stability enhancement achieved so far.

### 3.5. Effect of Glycosylation on the Solubility of EGCG

One of the main advantages of glycosylating phenolic compounds is related to the increased solubility of the glycoconjugates and, subsequently, to the improvement of their bioavailability in aqueous solutions [36,37]. As can be seen in Table 2, the solubility of the EGCG-glucoside, EGCG-sophoroside, and EGCG-xyloside was 44-, 23-, and 33-fold higher than that of EGCG, respectively. The data obtained in this study also agreed with previous results. Moon et al. synthesized three different α-glucosides, EGCG-G1 (glycosylated in the 7 position), EGCG-G2A (glycosylated in the 4′ position), and EGCG-G2B (glycosylated in the 4′ and the 7 position), and reported an improvement in solubility of 69-, 126-, and 122-fold, respectively [11]. Moreover, Zhang et al. [25] synthesized two β-glycosylated derivatives of EGCG (named EGCG-G1 (4″ position) and EGCG-G2 (4′ and 4″ position)) that were 15 and 31 times more soluble than the original compound. In summary, glycosylation of EGCG is a reliable strategy to augment the solubility of this molecule, although the effect that both the glycosylation site and the type of carbohydrate have on EGCG solubility needs to be studied further.

### 3.6. Antiproliferative Activity of EGCG Glycosides on Cancer Cells

EGCG has been reported to have an impact on different hallmarks of cancer, which makes it a promising molecule for future antitumor studies [38,39]. In particular, its potential in breast cancer is well documented [40,41], as it induces apoptosis in tumor cells. Considering the enormous antioxidant capacity of EGCG, it is interesting to notice that many human cancer cells exhibit a highly oxidative state due to decreased antioxidant levels of protective enzymes compared to those found in healthy tissues. Therefore, the addition of a highly antioxidant molecule can lead to the elimination of this high oxidative state, affecting the viability of these cells and maintaining non-transformed cells in a normal state [42].

In this work, we analyzed whether the glycosylation of EGCG affects its antiproliferative activity, using the human breast cancer cell line MDA-MB-231 as a tumor model, and the human non-transformed mammary epithelial cell line MCF10-A as a model of healthy mammary tissue.

Results from crystal violet assays revealed that, while EGCG did not affect the viability of cancer cells, the three glycosylated derivatives produced a concentration-dependent significant decrease (Figure 5A). However, we did not observe the antiproliferative response of EGCG on cancer cell cultures reported in the literature [25]. Most probably, this apparent discrepancy may be due to the fact that we measured cell viability 24 h after the EGCG challenge, while in the mentioned work, it was determined 48 and 72 h later.

Regarding the effect of the three derivatives, EGCG-sophoroside was the compound that induced the highest reduction in cell viability (about 60%), although the xyloside and glucoside derivatives also produced a significant decrease (40 and 25%, respectively). Zhang et al. [25] reported a reduction in breast cancer cell viability using 4″ and 4″ + 4′ combined EGCG glycosides, but increasing concentrations did not follow a dose–response behavior. Our data suggest that the 5″ hydroxyl group may be an interesting target to enhance the bioactive properties of EGCG. Moreover, the role of a sophorose derivative as an antiproliferative agent has been studied for the first time. The positive effect of this compound opens a study horizon for its possible use in other biomedical applications. Although it has been reported that sophorolipids, a family of glycolipid biosurfactants, could exhibit some antiproliferative effect [43,44], the role of sophorose itself needs further investigation.

In addition, to evaluate the safety profile of EGCG and the three derivatives, we analyzed their effect on the viability of the non-transformed MCF10-A cell line (Figure 5B). No changes in cell viability were observed upon treatment with any of the compounds tested, indicating their lack of cytotoxic effects on non-cancerous cells and their potential as an anticancer treatment. Overall, these results suggest that the glycosides, especially the EGCG-sophoroside, have better antiproliferative activity than EGCG on cancer cells and that all of them have a safe profile in non-tumor breast cells. Nevertheless, further studies are needed to understand the molecular events underlying cytotoxicity triggered by EGCG derivatives in the MDA-MB-231 cancer cell line.

### 3.7. Evaluation of the Neuroprotective Activity of EGCG Glycosides on Neuroblastoma Cell Lines

Neuroinflammation and oxidative stresses have been usually considered important factors in different neural illnesses, like Alzheimer’s disease, playing a crucial role in neurotoxicity. Thus, the use of antioxidant drug candidates may improve the treatment and prognosis of neurodegenerative diseases [45]. Several studies support the neuroprotective abilities of polyphenolic compounds like resveratrol, curcumin, quercetin [46], and even EGCG [47]. Furthermore, many studies regarding the potential effects of EGCG in Alzheimer’s disease (AD) treatment have been reported in the literature [48].

In the present work, preliminary assays about the neuroprotective activity of EGCG and the synthesized glycoconjugates were done using a cell model of tauopathies, such as AD, using the human neuroblastoma SH-S5Y5 cell line and the neurotoxicity induced by okadaic acid. Increasing evidence has suggested that inhibition of phosphatases by OA represents the most robust way to induce tau hyperphosphorylation. Thus, OA-treated cell lines have been used as established cellular models of hyperphosphorylated tau-induced neurodegeneration [49]. Firstly, the toxicity of EGCG and its glycosides on SH-S5Y5 cells was tested at a concentration range of 75–100 µM (Figure 6A). The results showed that incubating the cells in the presence of the EGCG and its derivatives not only did not decrease cell viability but slightly enhanced it.

Then, the neuroprotective activity of EGCG and its derivatives was tested in the presence of OA, a known inhibitor of protein phosphatases (Figure 6B). It is easy to observe the cell death produced by OA and how the treatment with EGCG and its glycosides produces a positive neuroprotective effect on cells. In general, all the molecules showed a dose–dependent behavior. Several compound concentrations induce significant neuroprotection, and the best results were obtained at the highest concentration tested (100 µM).

Under these conditions, both EGCG and the xylosylated derivative increased cell viability by approximately 40–50%, compared with cells treated with only OA. However, the most promising results were found when testing both glucosylated derivatives, especially the sophoroside. The neuroprotective effects of 100 µM concentrations of these glycosylated molecules significantly increased cell viability compared to control levels (75–95%). Few studies have tested the impact of glycosylated derivatives of EGCG in neuroprotection, but a similar increase in neuron viability was recently reported using 3′ and 7 α-glucosides [14].

## 4. Conclusions

Two-engineered GHs from the ascomycetous fungus *T. amestolkiae* were used to optimize the production of three novel EGCG glycosides by transglycosylation. These new catalysts showed their potential to glycosylate this bioactive substrate through a mechanism different from the one traditionally used for the synthesis of α-glucosides. Our results indicate that although the position of the glycosidic bond seems to be very important for antioxidant activity (EGCG glucoside and sophoroside shared similar antioxidant properties while the xyloside was less active), the solubility is more influenced by the identity or the length of the sugar attached (glucoside > xyloside > sophoroside). Moreover, glycosylation seems to play an important role in their antiproliferative and neurodegenerative properties, being the EGCG sophoroside the molecule with higher bioactive properties. To conclude, this work lays the groundwork for future studies that analyze in depth the efficacy of glycosylated derivatives of EGCG in neuroprotection or antiproliferation and opens new horizons for the use of glycosylated phenolic compounds in different therapeutic applications.

## Figures and Tables

**Figure 1 antioxidants-11-01325-f001:**
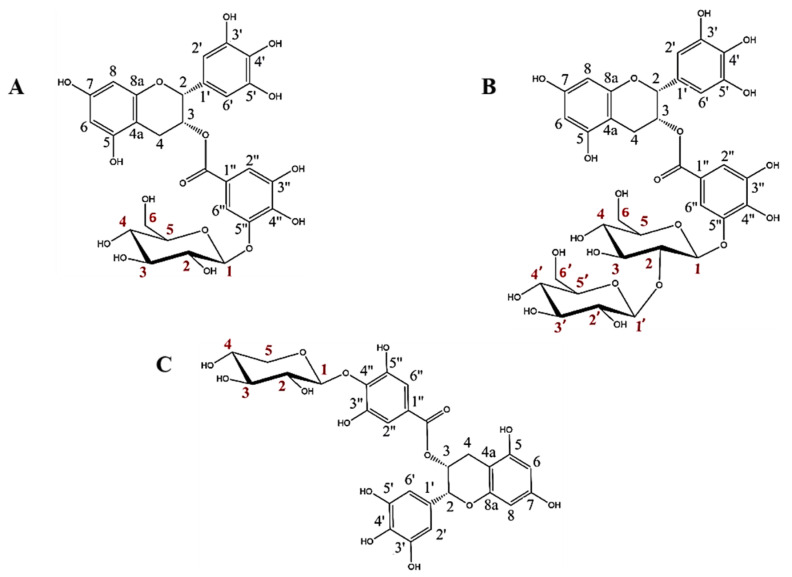
Structures of the three EGCG glycosides studied in this work: (**A**) epigallocatechin gallate 5”-O-β-d-glucopyranoside, (**B**) epigallocatechin gallate 5″-O-β-d-sophoropyranoside, and (**C**) epigallocatechin gallate 4″-O-β-d-xylopyranoside (adapted from Méndez-Liter et al. [19] and Nieto-Domínguez et al. [18]). Every C atom in the molecules is numbered in black for EGCG and in red for the glucose, sophorose and xylose motifs.

**Figure 2 antioxidants-11-01325-f002:**
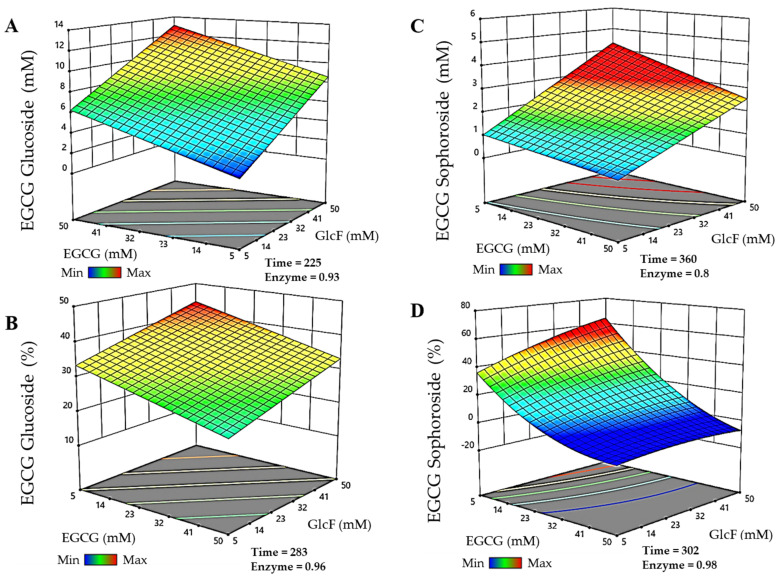
RSM prediction of the most efficient reactions catalyzed by the glycosynthase BGL-1-E521G, represented in three-dimensional surfaces. (**A**) Maximum production of epigallocatechin gallate 5″-O-β-d-glucopyranoside. (**B**) Maximum conversion of epigallocatechin gallate 5″-O-β-d-glucopyranoside. (**C**) Maximum production of epigallocatechin gallate 5″-O-β-d-sophoropyranoside. (**D**) Maximum conversion of epigallocatechin gallate 5″-O-β-d-sophoropyranoside. Time is represented in minutes. Enzyme concentration is given in g/L. The color code represents the production/conversion range from minimum (blue) to maximum (red).

**Figure 3 antioxidants-11-01325-f003:**
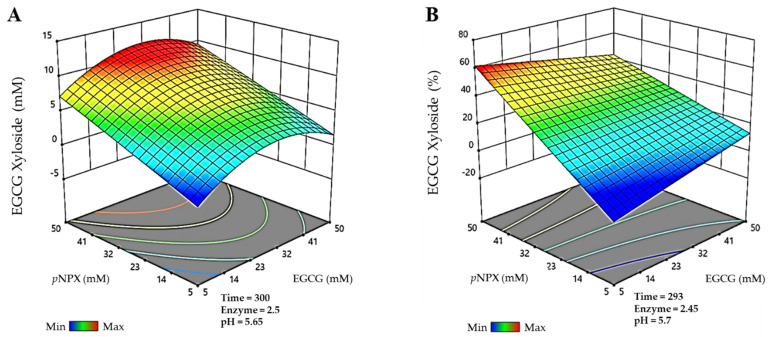
Prediction of the multiparametric model for the most efficient reactions catalyzed by the thioglycoligase BxTW1-E495A, represented in three-dimensional surfaces. (**A**) Maximum production of epigallocatechin gallate 4″-O-β-d-xylopyranoside. (**B**) Maximum conversion of epigallocatechin gallate 4″-O-β-d-xylopyranoside. Time is represented in minutes. Enzyme concentration is given in g/L. The color code represents the production/conversion range from minimum (blue) to maximum (red).

**Figure 4 antioxidants-11-01325-f004:**
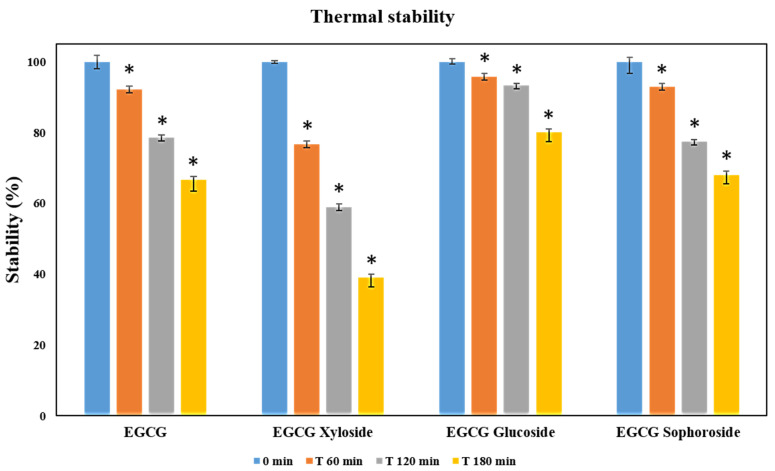
Thermal stability measured as relative degradation of EGCG and its glycosylated derivatives. In the study, 500 µM solutions of each molecule were incubated for 3 h at 70 °C. Student’s *t*-test: * *p* < 0.05 with respect to the 0 min measurement.

**Figure 5 antioxidants-11-01325-f005:**
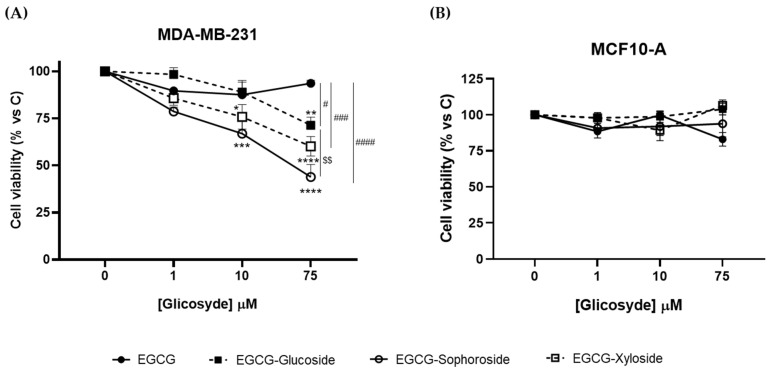
Effect of EGCG and its glycosides on the viability of (**A**) MDA-MB-231 and (**B**) MCF-10A cells after 24 h of compound addition. Cell viability was determined by crystal violet assay. Data represent mean  ±  SEM of three independent experiments. * *p*  <  0.05, ** *p*  <  0.01, *** *p*  <  0.005, **** *p*  <  0.001 vs. vehicle-treated cells. # *p*  <  0.05, ### *p*  <  0.005, #### *p*  <  0.001 vs. 75 μM EGCG-treated cells. $$ *p*  <  0.01 vs. 75 μM EGCG-Glucoside-treated cells.

**Figure 6 antioxidants-11-01325-f006:**
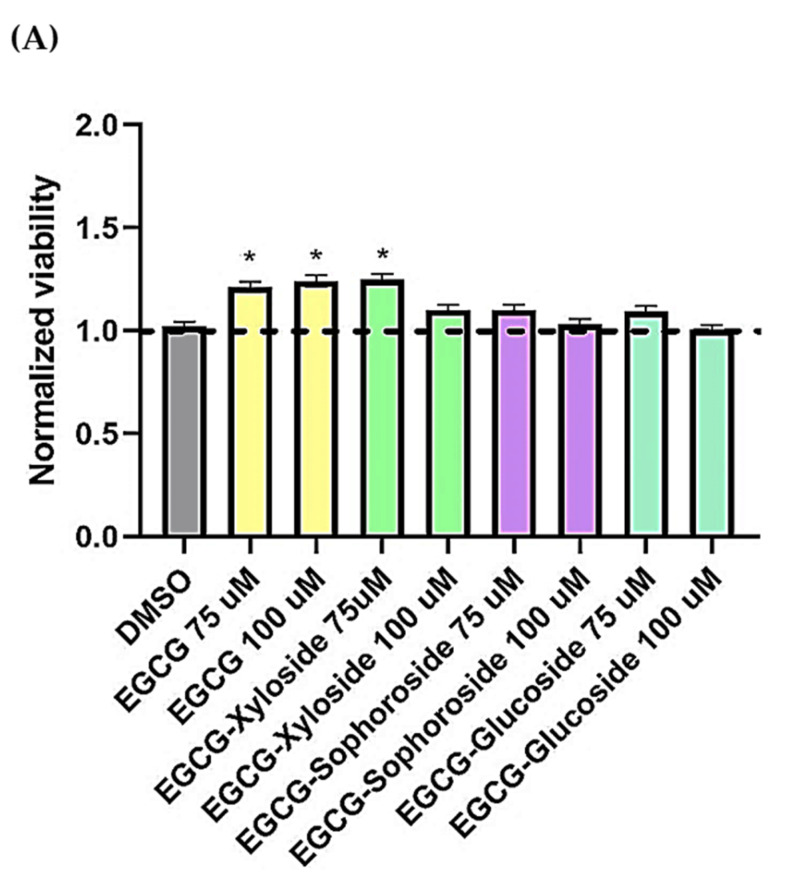
(**A**) Cell viability of SH-SY5Y cells after exposure to EGCG and its glycosides (75 and 100 μM). An MTT assay was used to determine cell viability 24 h after cells treatment. Each data point represents the mean ± SEM of two replications in three different experiments. * *p* < 0.05 vs. vehicle-treated cells. (**B**) Neuroprotective effect of EGCG derivatives using the OA-induced neurodegeneration model. SH-SY5Y cells were treated for 1 h with the compounds, at the concentrations indicated, prior the addition of OA. The cells were then incubated for 24 h. Cell viability was measured by MTT test. Mean ± SEM of at least three independent experiments. * *p* < 0.1, ** *p* < 0.01, *** *p* < 0.001.

**Table 1 antioxidants-11-01325-t001:** Antioxidant activity of EGCG and its glycosides using ABTS^•+^ as substrate.

Compound	R^2^	TEAC
Trolox	0.993	1
EGCG	0.992	0.135 ± 0.003
EGCG glucoside (5″)	0.994	0.180 ± 0.009 *
EGCG sophoroside (5″)	0.994	0.173 ± 0.012 *
EGCG xyloside (4″)	0.993	0.217 ± 0.016 *

Data expressed as mean ± SD (*n* = 4, * *p* < 0.01 vs. EGCG).

**Table 2 antioxidants-11-01325-t002:** Solubility determined for EGCG and its glycosylated derivatives.

Compound	Concentration (mM)	Solubility Increase (Fold)
EGCG	53.20	1.00
EGCG-Glucoside (5″)	2332.54	43.84
EGCG –Sophoroside (5″)	1220.37	22.94
EGCG-Xyloside (4″)	1776.98	33.40

## Data Availability

Data is contained within the article and Appendix A.

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
