# Peer review of "Glycosylation of Epigallocatechin Gallate by Engineered Glycoside Hydrolases from Talaromyces amestolkiae: Potential Antiproliferative and Neuroprotective Effect of These Molecules"

_antioxidants, 2022, doi:10.3390/antiox11071325_

Round 1

Reviewer 1 Report

The article is well written with clear methodology and results. However language edits and spell checks need to be performed. 

Author Response

We thank the positive opinion of reviewer 1 about our work. The English language has been revised.

Reviewer 2 Report

I have carefully examined the manuscript entitled “Glycosylation of epigallocatechin gallate by engineered glycosyde hydrolases from Talaromyces amestolkiae: antiproliferative and neuroprotective properties of the novel glycosides” by Méndez-Líter et al. Authors describe an efficient approach to synthesize three novel epigallocatechin gallate (EGCG) glycosides through glycosylation by engineered enzymes.

Authors performed an interesting and very accurate work, starting from a) the analysis of EGCG glycosylation and optimization of their enzymatic synthesis by response surface methodology; b) the impact of glycosylation on the solubility and thermal stability; c) the antioxidant, antiproliferative and neuro-protective activities of the new glycosylated EGCG’s. Results of their approach are remarkable and of high quality.

On this basis, I consider this work a useful contribution in the context of phenolic glycosides and their applications. Moreover, the manuscript doesn’t need language revision.

For these reasons I believe that the above-mentioned manuscript is suitable for publication on Antioxidants.

Author Response

We greatly appreciate the positive comments of reviewer 2.

Reviewer 3 Report

1. This study uses engineered glycosyde hydrolases from Talaromyces amestolkiae to produce three novel EGCG glycosides and then tests their abilities for anti-proliferation and neuroprotection. However, some major problems are found mainly in statistics and the lack of biological evidence.

2. All data should be analyzed by statistics and show the significant differences between means, including Figures 4, 5, 6, and Table 2.

3. The evidence for anti-proliferation and neuroprotection is too weak, the authors need to add more biological tests, such as Western blotting, the expression pattern of apoptosis-related genes, the expression pattern of genes involved in key signaling pathways of neuroprotection, etc., together with their photos of cells under microscopy observation to present the ability and morphology.

4. Altogether, this study looks too preliminary and should be added more experiments to support their conclusions.

5. L59: different length - different lengths

6. L78: an ascomycetous fungi - Do you mean an ascomycetous fungus?

7. L85: these glycoconjugates was optimized - these glycoconjugates were optimized

8. L212: is a very interesting approach to obtain - is a very interesting approach to obtaining

9. L215: has also been synthetized - has also been synthesized

10. Still many other grammar errors throughout the manuscript.

Author Response

  1. This study uses engineered glycosyde hydrolases from Talaromyces amestolkiae to produce three novel EGCG glycosides and then tests their abilities for anti-proliferation and neuroprotection. However, some major problems are found mainly in statistics and the lack of biological evidence.

We appreciate the effort of reviewer 3 in order to improve the quality of our work. A revision of the English language has been carried out and we have also introduced the comments and corrections mentioned by this reviewer.

  1. All data should be analyzed by statistics and show the significant differences between means, including Figures 4, 5, 6, and Table 2.

Significant differences were calculated for figures 5 and 6 but they were not highlighted. Sorry for this mistake. Some text and asterisks have been added to the figures to clarify it.

Regarding figure 4, two more replicates were carried out, and added in order to indicate the statistical analysis of the experiment. The figure was modified, and some text was added in M&M and results.

With respect to table 2, we should apologize, but the amount of glycosides that needs to be produced for reaching an N number of experiments that allow us to do the statistical analysis is not affordable. The amount of molecules required to generate a saturated solution, when working in a mM range, is very high. We are planning to scale up the production of these molecules in the frame work of a new proof concept Spanish project recently requested (https://www.aei.gob.es/en/announcements/announcements-finder/proyectos-prueba-concepto-2022).

  1. The evidence for anti-proliferation and neuroprotection is too weak, the authors need to add more biological tests, such as Western blotting, the expression pattern of apoptosis-related genes, the expression pattern of genes involved in key signaling pathways of neuroprotection, etc., together with their photos of cells under microscopy observation to present the ability and morphology.

We appreciate and understand the concerns of reviewer 3 about the biological activity tests of the glycosides. Although we agree with the comment, it is important to remark that the objective of the current work was to demonstrate the use of these novel catalysers as efficient tools to produce EGCG derivatives. The applications of the new glycosides in antiproliferative and neuroprotection applications is out of the scope of this work and will be studied in detail in a proof of concept Spanish project (state Program to Promote Scientific-Technical Research and its Transfer with financial support of EU). The data presented in this work suggest that glycosylated derivatives of EGCG may have the potential to be used as neuroprotective or antiproliferative agents, but these results are still far from a point where companies may feel interested in exploiting this knowledge. Continuation of this project will include scaling up the production of these compounds with optimization of the process from the beginning to the end, as well as a profound characterization of their antiproliferative and neuroprotective properties, with special emphasis in the description of their mechanism/s of action.

  1. Altogether, this study looks too preliminary and should be added more experiments to support their conclusions.

As previously noted, the objective of the current work was to prove the use of these novel catalysers as efficient tools to produce EGCG derivatives. The in depth study of the therapeutic potential of these products will be addressed in the frame work of the proof of concept project mentioned above

The indicated following mistakes have been corrected throughout the manuscript.

  1. L59: different length - different lengths
  2. L78: an ascomycetous fungi - Do you mean an ascomycetous fungus?
  3. L85: these glycoconjugates was optimized - these glycoconjugates were optimized
  4. L212: is a very interesting approach to obtain - is a very interesting approach to obtaining
  5. L215: has also been synthetized - has also been synthesized
  6. Still many other grammar errors throughout the manuscript.

Round 2

Reviewer 3 Report

Although statistics have been applied in the revised version, the biological tests for antiproliferative and neuroprotective properties are still lacking. The chief value of the two novel glycosides is their biological activities, just like it was mentioned in the title of this manuscript. I still feel this study is too basic and shallow and far below the average quality of this high-impact journal. Also, there is a risk to publishing such a kind of paper that only presents quantitative data without any biological tests and evidence such as Western blotting, PCR-based analysis and microscopic observation on cell viability and morphology using some molecular markers.

Author Response

As mention in the first round of the manuscript revision, we appreciate and understand the concerns of reviewer 3 about the biological activity tests of the glycosides. We agree with your comments, we would like to reiterate that the main objective of the current work was to improve the production of the three EGCG glycosides to demonstrate that our mutant enzymes are efficient glycosylation tools. In addition, we conducted preliminary studies to analyze the potential antiproliferative and neuroprotective actions of the resulting products, but it was completely out of the scope of this study to perform a detailed description of such effects. To make this clearer, we propose tuning down the title (Glycosylation of epigallocatechin gallate by engineered glycoside hydrolases from Talaromyces amestolkiae: potential antiproliferative and neuroprotective effect of these molecules), as well as other parts of the manuscript (highlighted in yellow).

However, if you consider that it is absolutely essential to add more data on the biological activity of the glycosides, we would have to request 30-40 days to perform experiments to show morphological changes of the cells in response to the compounds as well as the alteration of basic survival/death markers. Unfortunately, and as it was mentioned in our previous rebuttal letter, it will not be possible to perform additional experiments because we would need to scale up the production of the glycosides, which is not feasible at this time.